# Matrix Stiffness Modulates Metabolic Interaction between Human Stromal and Breast Cancer Cells to Stimulate Epithelial Motility

**DOI:** 10.3390/metabo11070432

**Published:** 2021-07-01

**Authors:** Iván Ponce, Nelson Garrido, Nicolás Tobar, Francisco Melo, Patricio C. Smith, Jorge Martínez

**Affiliations:** 1Cell Biology Laboratory, INTA, University of Chile, Santiago 7810000, Chile; ivponcebq@gmail.com (I.P.); nelson.garrido@usach.cl (N.G.); ntobar@inta.uchile.cl (N.T.); 2Physics Department, University of Santiago, Santiago 8320000, Chile; francisco.melo@usach.cl; 3Soft Matter Research and Technology Center, University of Santiago, Santiago 8320000, Chile; 4Department of Dentistry, Faculty of Medicine, Ponfificia Universidad Catolica de Chile, Santiago 8320000, Chile; psmithf@uc.cl

**Keywords:** stiffness, breast cancer, lactate, monocarboxylate transporters

## Abstract

Breast tumors belong to the type of desmoplastic lesion in which a stiffer tissue structure is a determinant of breast cancer progression and constitutes a risk factor for breast cancer development. It has been proposed that cancer-associated stromal cells (responsible for this fibrotic phenomenon) are able to metabolize glucose via lactate production, which supports the catabolic metabolism of cancer cells. The aim of this work was to investigate the possible functional link between these two processes. To measure the effect of matrix rigidity on metabolic determinations, we used compliant elastic polyacrylamide gels as a substrate material, to which matrix molecules were covalently linked. We evaluated metabolite transport in stromal cells using two different FRET (Fluorescence Resonance Energy Transfer) nanosensors specific for glucose and lactate. Cell migration/invasion was evaluated using Transwell devices. We show that increased stiffness stimulates lactate production and glucose uptake by mammary fibroblasts. This response was correlated with the expression of stromal glucose transporter Glut1 and monocarboxylate transporters MCT4. Moreover, mammary stromal cells cultured on stiff matrices generated soluble factors that stimulated epithelial breast migration in a stiffness-dependent manner. Using a normal breast stromal cell line, we found that a stiffer extracellular matrix favors the acquisition mechanistical properties that promote metabolic reprograming and also constitute a stimulus for epithelial motility. This new knowledge will help us to better understand the complex relationship between fibrosis, metabolic reprogramming, and cancer malignancy.

## 1. Introduction

Breast tumors belong to a type of epithelial neoplastic lesion characterized by a dense structure that contains mainly stromal cells and abundant extracellular matrix (ECM) molecules. Reciprocal interactions between these components and epithelial cells determine the course of cancer progression in a desmoplastic environment [1]. Desmoplastic tissue can have a heterogeneous composition ranging from a predominantly cellular stroma containing fibroblasts, vascular cells, and immune cells with little ECM, to a dense structure with minimal cells and low ECM content, consisting mainly of fibrillar collagen [2]. In the case of mammary tissue, a tumor stroma also exhibits a reduction in the size and number of adipocytes, the most abundant cell type of breast tissue [3].

It has been established that a stiffer tissue structure is a determinant of breast cancer progression. Provenzano and colleagues, using a mouse model of breast cancer, observed that higher stromal collagen density in mammary tissue resulted in a 3-fold increase in tumor number and that tumors displayed a more invasive phenotype with greater local invasion and metastasis [4]. Analyses of human breast cancer samples have revealed that tumor stiffness reflects a more aggressive disease. Particularly, authors have demonstrated that the transition from a non-malignant tissue to an invasive ductal carcinoma corresponds to significant collagen deposition, linearization, and bundling, which leads to a stiffening of the ECM [5].

For many years, desmoplasia was considered to be a reaction of host tissue against invasive cancer cells, which was designated “desmoplastic response” [6]. However, this idea has since been challenged using mammographic studies that identify desmoplastic tissue in normal human breast detected as areas denser than the predominant mammary fat tissue, characteristically designated as “mammary density” (MD). Radiodense areas exhibit several histologic characteristics of a malignant stroma, specifically low adipocyte content and high ECM and stromal cell abundancy [7]. The finding that desmoplastic lesions can be present in the absence of tumor cells suggests that desmoplasia may not be a reaction to invasive malignant cells, but, rather, a preexisting condition that creates a proactive milieu that favors the development of cancer [8].

It has also been proposed that an abundant stromal cell population plays an important role in the establishment of the metabolic reprogramming phenomenon characteristic of the majority of tumors [9].

The specific mechanisms that modulate this metabolic arrangement can be achieved either by the stromal or epithelial populations with a focus on the mechanisms of substrate transport. Lisanti and colleagues have proposed that tumor-associated fibroblasts (also termed as CAFs) establish a cooperative relationship with tumor cells where CAFs act as a glycolytic source of lactate which, in turn, can be used by cancer cells and oxidized in the mitochondria for energy production [10]. Thus, at the whole tumor level, an increased conversion of glucose to lactate, associated with a high glycolytic rate, generates millimolar concentrations of lactic acid, which is released to the extracellular compartment [11]. The significance of lactate for the physiology of the mammary environment is still poorly understood.

In the present work, using a normal breast stromal cell line, we present evidence that suggests that modulation of ECM dynamics and metabolic reprogramming could constitute mechanistically related processes. Our results show that a culture of mammary stromal cells on matrices of increased stiffness display an enhanced production of lactate that correlates with an increased uptake of glucose, an enhanced expression of the glucose transporter Glut1 and also a stimulus for the expression of membrane monocarboxylate transporters (MCT4) that facilitates lactate export. Moreover, mammary stromal cells cultured on stiffer matrices generate soluble factors, lactate among them, which stimulate epithelial breast migration in a stiffness-dependent manner. Our analysis is mainly focused on the stromal cells that, in desmoplastic tumors, like the breast lesions, correspond to the main component of tumor architecture and could be responsible for the metabolic rearrangement that characterizes these tumors.

## 2. Results

### 2.1. RMF-621 Cells in Culture Release Lactate in a Stiffness-Dependent Manner

To assess whether matrix stiffness modulates lactate production, we cultured RMF-621 cells for 3 days in media enriched with 10% FCS on a 0.2 kPa low-rigidity matrix (Advanced Biomatrix, San Diego, CA, USA) in an attempt to simulate the normal soft condition of breast tissue [5]. The cells were then transferred to a set of dishes with different elastic moduli of 0.2, 2, 16, and 36 kPa coated with type I collagen (50 µg/cm^2^) and incubated for 72 h. Afterwards, lactate production was evaluated in culture media by gas chromatography coupled with mass spectrometry (GC–MS), as detailed in Section 4. As Figure 1 shows, lactate production by stromal cells responds in a linear manner to matrix stiffness, which suggests that the mechanical signals derived from the matrix constitute a specific stimulus for glucose reprogramming.

### 2.2. Matrix Rigidity Favors Glucose Uptake in Mammary Stromal Cells

To assess whether matrix stiffness affects the ability of stromal cells to uptake glucose, we plated RMF-621 cells on coverslips coated with an elastic polyacrylamide gel that generates matrices with stiffnesses of 4.5 and 31 kPa, to which type I collagen was covalently cross-linked [12]. The cells were incubated for 3 days in this condition prior to being transduced (for 24 h) with an adenovirus that encoded the FRET glucose imaging biosensor FLIPglu-60060Δ6, which displays a high affinity for glucose [13]. As Figure 2 shows, cells seeded on a stiffer matrix exhibited consistently higher incorporation of glucose (double V initial value) that allows us to suggest that a rigid microenvironment can favor glucose uptake and incorporate substrate for metabolic reprogramming.

### 2.3. Matrix Stiffness Stimulates the Expression of Monocarboxylate and Glucose Transporters in Normal Stromal Mammary Cells

Next, we assessed whether a culture of RMF-621 cells on matrices with different stiffnesses was able to modulate the expression of transporters involved either in the uptake of glucose by Glut1 or the export of lactate by MCT4, a monocarboxylate transporter that appears to be the main mechanism adapted to the release of lactate [14]. Figure 3 shows a Western blot of both transporters expressed by RMF-621 cells cultured for 72 h on matrices of increased stiffness. As Figure 3 indicates, both Glut1 and MCT4 protein expression was stimulated by matrix stiffness. These results allow us to argue that stiffer conditions favor glucose uptake and lactate export.

### 2.4. Matrix Stiffness Modulates Lactate Transport in MDA MB-231 Cells

To investigate whether the modulation of monocarboxylate expression in epithelial cells by plating in different stiffness matrices is reflected in lactate transport, we performed lactate transport experiments using a genetically encoded lactate-sensitive FRET nanosensor Laconic [15]. It is important to note that regardless of the net direction of lactate flux, MCTs transport in both directions, and therefore, their activity may be studied by monitoring influx or efflux, the former being more practical. We used the same experimental approach for glucose transport, as depicted in Figure 2. Epithelial MDA MB-231 cells were plated (72 h) on coverslips coated with polyacrylamide and cross-linked with type I collagen. Figure 4 shows that cells plated on a 31 kPa matrix incorporate lactate at a higher rate than cells plated on a 4.5 kPa matrix.

### 2.5. Soluble Factors Derived from RMF-621 Cells Cultured in Different Levels of Stiffness Stimulate Epithelial Migration

To test whether stromal soluble factors secreted by RMF-621 cells cultured on a range of matrices with different rigidity can influence the migratory behavior of their epithelial counterpart, we designed the following experiments. Using MCF-7 cells (a weakly migratory mammary cell line) and MDA MB-231 (a highly invasive cell line), we performed migratory and invasion experiments using the Transwell system, in which epithelial cells migrated via attraction by the stimulus of 50% media conditioned by RMF-621 cells previously cultured on different matrices. This medium occupied the lower space of the Transwell chamber. To analyze whether lactate produced by these stromal cells plays a role in the migratory stimulus, both epithelial cells were allowed to migrate in the presence of 1 µM MCT1/2 inhibitor ARC155858 [16] and 1 mM diclofenac, a structurally-unrelated MCT1 and MCT4 blocker [17]. As Figure 5 shows, in both cell lines, epithelial cell migration responds to stiffness-derived stromal stimulus in a directly proportional way, that is, media conditioned from RMF-621 cells cultured on a more rigid matrix generate a more vigorous migratory stimulus. The figure also shows that both ARC155858 and diclofenac inhibit cell migration (to different degrees) in MCF-7 cells, suggesting that in these cells, lactate, incorporated via the abundantly expressed MCT1 transporter [18], was at least partially responsible for epithelial migration, at least in the higher stiffness levels. On the other hand, MDA MB-231 cell invasion was insensitive to ARC155858 and lactate-dependent invasion was blocked only by diclofenac. These results suggest that in our experimental conditions, MDA MB-231 cells incorporate lactate using MCT4, a monocarboxylate transporter predominantly expressed in these cells, the expression of which has been associated mainly with lactate export. [18,19].

### 2.6. MCT1 and MCT4 Epithelial Expression Associates with Breast Cancer Grading

It has been demonstrated that increased ECM rigidity correlates with cancer progression and poor disease outcome [20]. To verify whether the modulation of monocarboxylate transporters by matrix stiffness observed in our cellular model correlates with human tumor aggressiveness, we analyzed the expression of both transporters in fixed samples of human breast tissue derived from tumor-bearing patients with different Elston grades [21] using tissue immunofluorescence. As Figure 6 shows, the Elston Grade I sample showed a strong expression of MCT1 in the almost intact epithelial compartment, with low expression in the stroma. On the contrary, the Grade III sample displayed a strong reactivity to MCT4, associated with the surface of epithelial cells and a weak stain of MCT1 in both compartments. These results are in agreement with the data in Figure 5 that MDA MB-231 cells, an aggressive cell line, also express a high abundance of MCT4 [18].

## 3. Discussion

Glucose reprogramming and deregulated ECM dynamics act as fundamental drivers of cancer progression and can be interconnected and act in a cooperative manner [22]. The results of the present work support the notion that matrix stiffness constitutes a specific stimulus for glucose uptake modulation, lactate production, and the expression of monocarboxylate and glucose transporters by stromal cells. In the same experimental conditions, stromal cells generate soluble factors that stimulate epithelial migration.

It has been proposed that changes in tensional forces and ECM stiffness are determinant features in cancer progression [23]. Moreover, tissues that are normally stiff exhibit a greater risk of developing cancer than softer tissues with similar division rates [24]. In the case of breast cancer, mammographic studies have identified the existence of desmoplastic areas that exhibit histologic characteristics of malignant stroma due to the increased proportion of collagen relative to fat tissue content [7]. The finding that desmoplastic lesions can be present with the absence of tumor cells suggests that desmoplasia may be a preexisting condition favoring the development of cancer [8].

Studies using metformin, an agonist of AMP-activated protein kinase (AMPK), have shown a functional relationship between metabolism and fibrosis. Using a mouse model of cardiac fibrosis, authors have demonstrated that metformin reduces levels of TGF-β and activated Smad3, thus decreasing collagen deposition and fibrosis [25]. More recently, it has been shown that a switch of metabolism from oxidative phosphorylation to aerobic glycolysis (Warburg effect) in renal fibroblasts was the primary feature of fibroblast activation during renal fibrosis [26].

In the present work, we tested the hypothesis that a pre-established matrix with increased stiffness constituted a primary mechanical stimulus that favors glucose reprogramming. Previous data from our laboratory show that mammary fibroblasts behave as lactate producer cells and epithelial cells are avid consumers of this substrate, a property that directly depends on cell malignancy [18]. When mammary stromal cells were exposed to a stiffer matrix, their intrinsic lactate production was stimulated in a rigidity-dependent manner. This result, besides those obtained in glucose uptake experiments, allows us to suggest that a combination of surface topography and substrate stiffness exert some type of intracellular forces that promote, in the stromal compartment, the expression of regulatory proteins that favor a more efficient uptake of glucose and an enhancement of lactate extrusion. Both phenomena are key elements in the metabolic demand of epithelia in transformation. The correlation between matrix rigidity and glucose metabolism in the context of malignancy has been suggested by works performed in human prostatic samples. Results show that the mRNA expression of lysyl oxidase (LOX) and GLUT-1, proteins involved in the crosslinking of collagen and glucose transporters, correlate well with an increased cancer grade evaluated by the Gleason score [27].

We propose that mammary stromal cells are able to sense the stimulus of matrices of increased stiffness and convert this exogenous signal to an intracellular one, that is, lactate production, which stimulates epithelial motility. Thus, in the context of the tumor environment, lactate expands this functional repertoire from a relevant agent of the Warburg effect to a specific stimulator of the intracellular locomotion machinery. Interestingly, malignant breast cancer cells seem to be adapted to respond to lactate stimulus by the abundant expression of MCT4. This is demonstrated in the MDA MB-231 cell line, which corresponds to a triple-negative (TN) isotype, characterized by its aggressiveness [28], and of which we previously reported a high avidity for lactate correlates with the expression of abundant copies of MCT4 and high Km for lactate [16]. It has been previously proposed that MCT4, by its interaction with integrin β1, plays a relevant role in epithelial cell migration [29]. Moreover, MCT4 is the only monocarboxylate transporter through which expression is modulated (via HIF-1α) by a hypoxic environment, a feature characteristic of malignant tumors [30]. To date, no specific inhibitors for MCT4 have been described. Despite this, diclofenac (a well-known nonsteroidal anti-inflammatory drug) behaves as a strong, non-competitive inhibitor of monocarboxylate transport [31]. Recently, it has been proposed that MCT4 can be a potential therapeutic target in leukemic patients and that some of its inhibitors can serve as a new anti-proliferative drug in combination with conventional chemotherapeutic agents [32]. Our data reinforce the possible association of MCT4 and malignancy in breast cancer by the observation from patient samples that the expression of this MCT4 was also differentially expressed in the epithelial compartment of tumors of a different Elston Grade (Figure 6).

Together, the results shown in the present work propose that matrix stiffness constitutes not only a scaffold where a tumor can develop but also an early signal that promotes a shift of metabolic behavior of the stromal compartment through a glycolytic process that results in a stimulus of tumor malignancy. These new findings will help us to build a more complete picture of the relationship between fibrosis and cancer malignancy.

## 4. Materials and Methods

### 4.1. Cell Culture, Cell Lines, and Chemicals

We used human cell line RMF-621, which corresponds to hTERT-immortalized mammary fibroblasts derived from a reduction mammoplasty obtained via a generous gift from Dr. Charlotte Kuperwasser (Tufts University, Boston MA, USA) [33]. The RMF-621 cells were cultured in high glucose (25 mM) Dulbecco’s modified Eagle’s medium (DMEM) (Invitrogen, Carlsbad, CA, USA), supplemented with antibiotics and 10% fetal bovine serum (FBS) (Hyclone, Logan, UT, USA) and maintained in a humidified atmosphere of 37 °C, 5% CO2. MCF-7 and MDA MB-231, both human epithelial breast adenocarcinoma cell lines, were purchased from ATCC (Manassas, VA, USA), cultured in DMEM/F12 (17.5 mM glucose), supplemented with 10% FBS, and maintained in the same conditions as above. MCT1 inhibitor AR-C155858 was acquired from Chem Express (Princeton, NJ, USA), and diclofenac was purchased from Sigma (St. Louis, MO, USA).

### 4.2. Plating of Cells in Matrices of Different Stiffnesses

In all the experiments in which RMF-621 cells were cultured in matrices with different stiffnesses, cells (2 × 10^5^) were pre-cultured for 3 days in media enriched with 10% FCS plated on a 0.2 kPa low-rigidity matrix (Advanced Biomatrix, San Diego, CA, USA) previously coated with 50 μg/cm^2^ of type I collagen to simulate the normal (soft) condition of breast tissue. After this, the cells were released with trypsin and transferred to plates with increased stiffness (0.2–32 kPa) for an additional 72 h period before being lysed for Western blot analysis. In the experiments in which media conditioned from these cultures were collected, the enriched culture media was replaced by serum-free media for the last 24 h of culture before collection.

### 4.3. Preparation of Protein-Coated Gel Substrate

To measure the effect of ECM rigidity on glucose and lactate transport, we used compliant elastic polyacrylamide gels as a substrate material to which ECM molecules can be covalently linked [12]. For these studies, preactivated coverslips were used to prepare the matrices, which were cross-linked with 2 mM sulfo-SAMPAH and activated with 7500 J of UV light for 8 min. After rinsing with PBS, the matrices were incubated overnight at 4 °C with 50 μg/cm^2^ of type I collagen (Advanced Biomatrix, San Diego, CA, USA), and then washed with PBS. Once the collagen was coupled, the matrices on the coverslips were rinsed with sterile PBS and the cells were seeded on top.

### 4.4. Cell Imaging

To evaluate glucose transport in cells seeded on coverslips coated with extracellular matrices with different stiffnesses, we used the FRET nanosensor (FLIIPglu60060Δ6) incorporated with adenoviral particles (Vector Biolabs, Philadelphia, PA, USA), which exploit resonance energy transferred between a coupled pair of cyan and yellow fluorescent proteins (eCFP, eYFP). This system detects conformational changes induced by a glucose-binding domain derived from chemotactic receptors of bacteria, namely the glucose/galactose-binding protein of Escherichia coli (MglB) [13].

To evaluate lactate transport in living cells, we used the FRET nanosensor Laconic [15] incorporated with adenoviral particles (Vector Biolabs, Philadelphia, PA, USA). Laconic is a fusion protein composed of a ligand-binding moiety (LldR) specific to lactate and the fluorescent proteins mTFP and Venus. To evaluate the lactate uptake capacity of MDA-MB231 cells, we performed a zero-trans protocol using 5 mM lactate in the extracellular space [18]. A biosensor signal was evaluated by imaging recording using a Nikon Ti microscope with a 40× objective equipped with a monochromator (Cairn Research, Kent, UK), which allows discrete excitation at 430 ± 10 nm. Two windows of emitted light were simultaneously collected at 490–520 nm (mTFP) and over 535 nm (Venus) by means of an optical splitter (Cairn Research, Kent, UK). The images were digitized by a CCD camera (ORCA3, Hamamatsu, Japan), and the data were expressed as the ratio between mTFP and Venus fluorescence. The experiments were conducted 48–72 h after infection at room temperature (24–26 °C) in KRH buffer (in mM: 140 NaCl, 4.7 KCl, 20 Hepes, 1.25 MgSO_4_, 1.25 CaCl_2_, pH 7.4) supplemented with lactate.

### 4.5. Cell Motility Assays

Low migrating MCF-7 and highly invasive MDA MB-231 mammary epithelial cell lines were used as models of differential migration and invasion using a 6.5-mm Transwell chamber with a pore size of 8 µm (Corning, Corning, NY, USA). In these experiments, 5 × 10^4^ epithelial cells were allowed to migrate for 24 h (MCF-7) and invade for 16 h (MDA MB-231). In the case of the latter, cell invasion was studied using the same system but with coating the Transwell membrane with 10 mg/mL of Matrigel on the topside. Both epithelial cells migrated to the stimulus of 50% conditioned medium generated by RMF-621 cells previously cultured as described above, with increased stiffness conditions (0.2, 2.0, 16, and 32 kPa), enriched with 1% FCS in culture media. To analyze the effect of stromal-derived lactate on epithelial migration and the role of MCT1 and MCT4 monocarboxylate transporters, groups of both cells lines were stimulated to migrate under the stimulus of stromal conditioned media in the presence of 1 µM AR-C155858 (inhibitor of MCT1 and MCT2) or 1 mM diclofenac a structurally-unrelated MCT1 and MCT4 blocker. After the migration period, the Transwell membrane was fixed in methanol and migratory cells were stained on the lower side of the membrane with 0.2% crystal violet [34]. The migration values correspond to the average of 3 independent experiments by counting 16 fields from 4 pictures (×20) per chamber (2 chambers per experimental condition).

### 4.6. Lactate Assay

Lactate was evaluated in conditioned media by RMF-621 cells (1 × 10^5^ cells in a 6 well plate) cultured on matrices of increasing stiffness. To do so, we previously cultured RMF-621 cells on a 0.2 kPa low-rigidity matrix as described and then the cells were transferred to a set of dishes coated with matrices of increasing stiffnesses of 0.2, 2, 16, and 36 kPa. Lactate production was evaluated after 72 h of culture on these matrices of different stiffness levels. In the last 4 h of culture, the culture media was changed to phenol red-free media in the absence of serum, and lactate abundance in these media was evaluated by gas chromatography coupled with mass spectrometry, the GC-MS equipped with a capillary column HP5M. Briefly, an internal standard was added to each sample and the organic acid was oximated and extracted twice with ethyl acetate as previously described [35]. The organic fraction obtained was dried under nitrogen and derivatized with BSTFA (N,O-Bis(trimethylsilyl)trifluoroacetamide) and TMS (N-ethynyl-N,4-dimethylbenzenesulfonamide) (1%). Afterwards, one microliter of the derivatized sample was injected into the GC-MS. Each metabolite was identified based on its own mass spectra by matching with a spectral library of known metabolites from NIST, (National Institute of Standars and Technoligy). The quantification of lactic and pyruvic acids was carried out through prior elaboration of a calibration curve (quadratic equation) with increasing amounts of each metabolite normalized to Tropic Acid (Internal Standard).

### 4.7. Matrix Stiffness Measurements

To measure the elastic modulus of matrices used as cell supports in the imaging experiment, an indentation method was implemented. A glass sphere that is pushed against the matrix at constant speed was selected to ensure a defined geometry for indentation. The compression force was determined through a load cell, Futek LBS200 S-Beam, operating in the range of 2 N, which was provided with a signal-conditioning module and the respective interface for the conversion and transmission of data to the computer through the USB computer port. The sample (1 mm thick and 1 cm in diameter) was positioned horizontally under the spherical indenter (5 mm in diameter). To press the sample, the indenter was displaced by means of a micro-control system, with a 1 µm resolution, at a speed of 10 µm/s.

The resulting force, F, as a function of the penetration distance, δ, is modeled by the Hertz’s relation, $F=\frac{4}{3}{\;E}^{*}{\;R}^{1/2}{\;\delta }^{3/2}$, with $E^{*}=E/(1-\nu^{2}$), where E is the Young’s modulus and **ν** is the Poisson’s coefficient of the sample, which is assumed to be close to ν = 0.5. R is the indenter’s radius of curvature at the sample contact. E is assessed through a standard fitting procedure of the F vs. δ curve that is well represented by the Hertz’s formula. The values obtained for the substrata prepared at two distinct reticulation periods were as follows:

Sample 1: E = 4.5 ± 0.5 kPa

Sample 2: E = 31.0 ± 0.5 kPa

All force measurements were obtained from samples immersed in PBS buffer under the same conditions as those of the cell cultures. 

### 4.8. Western Blot and Antibodies

The expression of MCT4 and Glut 1 protein in RMF-621 cells was evaluated by Western blot. Briefly, cells previously plated for 72 h on matrices of increasing stiffness (0.2, 2, 16, and 36 kPa) were lysed in lysis buffer (30 mM Tris-HCl pH 7.5, 5.0 mM EDTA, 150 mM NaCl, 1% Triton X-100, 0.5% sodium deoxycolate, 0.1% SDS, and 10% glycerol) supplemented with complete protease inhibitors (Roche, Mannheim, Germany). Pellets were incubated for 1 h in lysis buffer at 4 °C and then centrifuged at 14,000× *g* for 15 m at 4 °C, keeping the supernatants. The protein concentration of cell lysates was determined using a Pierce BCA Protein Assay kit (Thermo, Rockford, IL, USA). Protein extracts were denatured in sodium dodecyl sulfate (SDS)–polyacrylamide gel electrophoresis loading buffer 4× (240 mM Tris–HCl, pH 6.8, 8% SDS, 40% glycerol, and 20% 2-mercaptoethanol), incubating the samples for 1 h at 37 °C. Equal amounts of protein from different treatments were resolved by SDS–polyacrylamide gel electrophoresis in 10% acrylamide gels and electrotransferred to polyvinylidene difluoride membranes using a buffer containing 24 mM Tris, 194 mM glycine, and 20% methanol. The proteins were further analyzed using the Supersignal West Dura Extended Duration Substrate (Thermo, Rockford, IL, USA). Immunoreactions were achieved by incubation of the membranes, previously blocked with a solution containing 5% bovine serum albumin in Tris-buffered saline and 0.05% Tween 20 (Sigma, St. Louis, MO, USA), with anti MCT4 (D-1) mouse monoclonal antibody (sc-376140) from Santa Cruz (Santa Cruz, CA, USA), anti-MCT1 rabbit polyclonal antibody (M4470-01B) from US Biological (Salem, MA, USA), anti-Glut1 mouse monoclonal antibody (MAB14181) from R & D (Minneapolis, MN, USA) and mouse anti-alpha tubulin (T5168) from Sigma (St. Louis, MO, USA). Densitometric analysis of Western blot bands was performed using a C-Digit Blot Scanner and Image Studio Digits software v.5.2 from LI-COR Biosciences (Lincoln, NE, USA).

### 4.9. Human Breast Cancer Samples

Breast cancer formalin-fixed samples and paraffin-embedded samples were obtained from the repository of the Arturo Lopez Perez Foundation (FALP, Santiago, Chile). Informed consent was obtained from all individual participants included in the study with the supervision and approval of the Ethics Committees at INTA, University of Chile and FALP. The samples were obtained before the beginning of chemotherapy.

### 4.10. Statistical Analysis

Unless otherwise specified, the experiments were repeated between three and five times. All the data are presented as means ± standard error of the mean (SEM). Statistical analyses were carried out using the GraphPad Prism software version 8.0.2 (GraphPad Software Inc. La Jolla CA, USA). Differences between groups were evaluated with the Kruskal–Wallis or Friedman analysis, followed by Dunn’s multiple comparisons test. A *p*-value lower than or equal to 0.05 was considered statistically significant and indicated with one, two or three asterisks: (*) = *p* ≤ *0*.05, (**) = *p* ≤ *0*.01 and (***) = *p* ≤ *0*.001.

## Figures and Tables

**Figure 1 metabolites-11-00432-f001:**
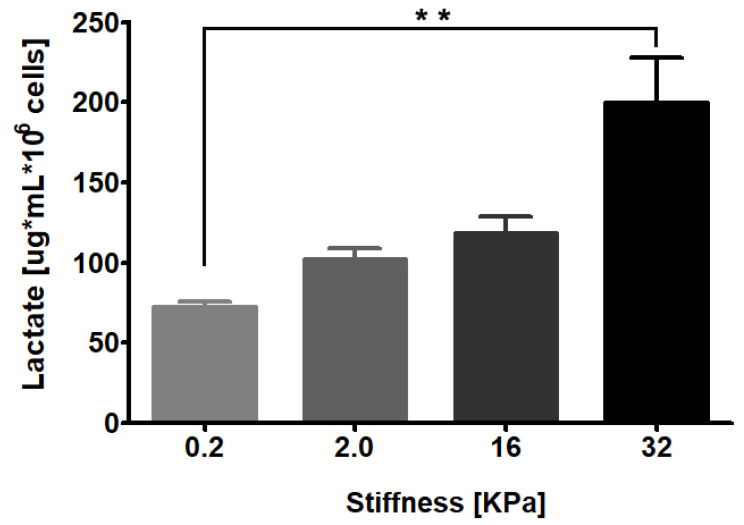
RMF-621 cells behave as lactate producer cells in a stiffness-dependent manner. Mammary human stromal cells (RMF-621 cells) were previously cultured (3 days) in media enriched with 10% FCS on a 0.2 kPa low-rigidity matrix (Advanced Biomatrix) to simulate the normal soft condition of breast tissue. After this, the cells were transferred to a set of matrices with increasing stiffness (0.2, 2.0, 16, and 36 kPa) for 72 h. Further, the lactate present in serum-free medium was evaluated, as explained in Section 4. The data are presented as the means ± standard error (SEM). (**) indicates statistically significant differences, with *p* ≤ 0.01 using Kruskal–Wallis test, followed by Dunn’s multiple comparisons test.

**Figure 2 metabolites-11-00432-f002:**
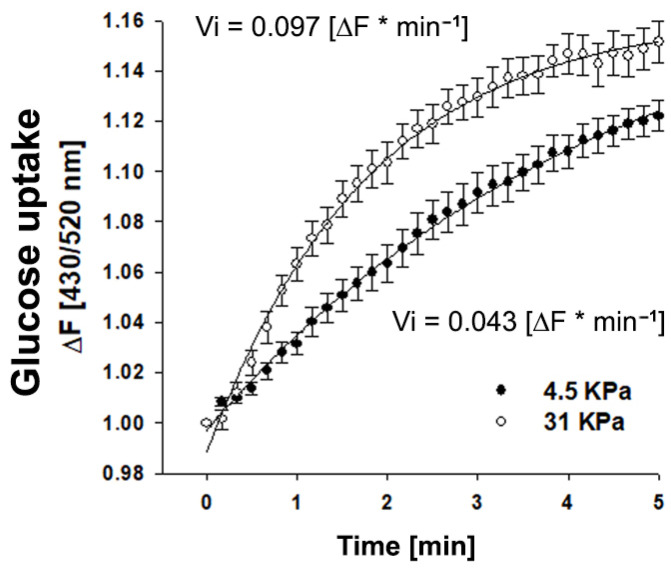
Glucose uptake is favored in mammary stromal cells cultured in rigid matrices. Time course curve of glucose incorporation measured as the fluorescence ratio of FLIPglu-60060Δ6 biosensor in RMF-621 cells cultivated for 4 days on coverslips coated with an elastic polyacrylamide gel, to which type I collagen (50 µg/cm^2^) was covalently cross-linked and exposed to 5 mM glucose. Fluorescence was recorded every 10 sec. Open circles correspond to cells cultured on a soft matrix (4.5 KPa), and closed circles to cells cultured on a stiffer matrix (31 KPa). The data correspond to an average of almost 20 single cells recorded in four independent experiments. The continuous line represents the best fit of a double rectangular hyperbola, and the Vi values were estimated from the slope of each glucose uptake curve.

**Figure 3 metabolites-11-00432-f003:**
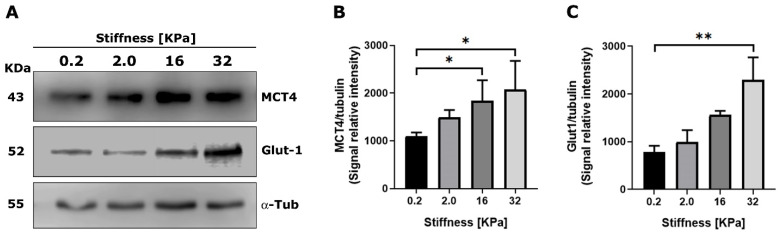
Extracellular matrix stiffness modulates the expression of monocarboxylate transporters in mammary stromal cells. RMF 621 cells (2.5 × 10^5^ per plate) were incubated on plates displaying increasing stiffness and coated with collagen I, as described above, in media +10% FCS. Afterward, 72 cells were lysed and Western blots of MCT4 (**A**) and Glut1 (**B**) were performed as described in Section 4. Densitometric analyses of three different experiments were performed from three independent experiments, as shown in (**C**). The data are presented as the means ± standard error (SEM). (*) and (**) indicate statistically significant differences, with *p* ≤ 0.05 and *p* ≤ 0.01, respectively, using the Friedman test followed by Dunn’s posthoc test.

**Figure 4 metabolites-11-00432-f004:**
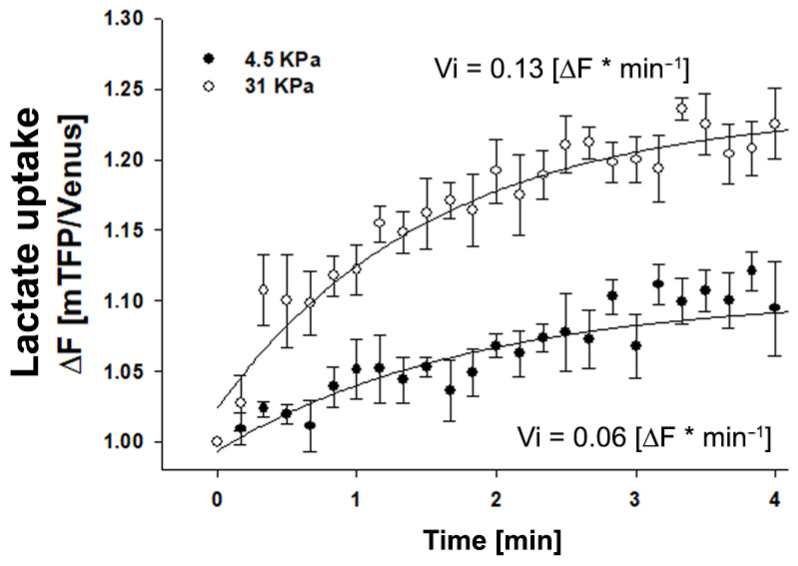
Lactate transport is favored in mammary MDA MB-231 cells cultured in rigid matrices. Time course curve of lactate incorporation in MDA MB-231 cells measured as the fluorescence ratio of a LACONIC biosensor. Cells were cultivated for 4 days on coverslips coated with an elastic polyacrylamide gel to which type I collagen was covalently cross-linked, and then exposed to 5 mM lactate. Fluorescence was recorded every 10 sec. Open circles correspond to cells cultured on a soft matrix (4.5 KPa), and closed circles to cells cultured on a stiffer matrix (31 KPa). The data correspond to an average of three single cells recorded in two independent experiments. The continuous line represents the best fit of a double rectangular hyperbola, and Vi values were estimated from the slope of each lactate uptake curve.

**Figure 5 metabolites-11-00432-f005:**
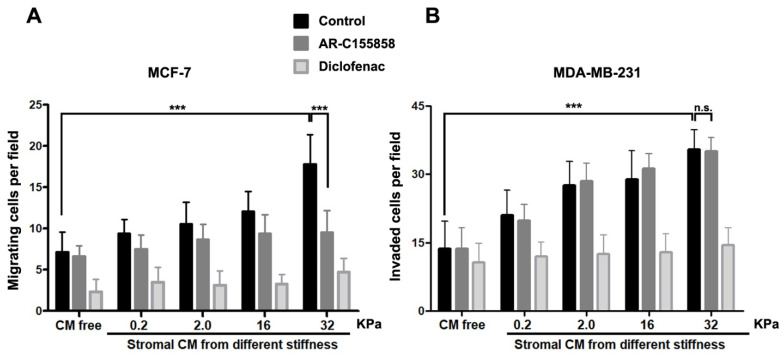
Epithelial migration was stimulated by media conditioned by stromal cells previously plated on increasing stiffness surfaces. The differential role of monocarboxylate transporters. MCF-7 and MDA MB-231 epithelial cells (5 × 10^4^) were stimulated to migrate in a Transwell system for 24 and 16 h, respectively, under the stimulus of 50% media conditioned by RMF 621 cells previously plated on surfaces with increasing stiffness. To test the role of MCTs in this stimulus, in a group of cells, migration was performed in the presence of 1 µM of AR–C155858 (which blocks MCT1 and MCT2 but not MCT4) and in another, in the presence of 1 mM of diclofenac (a structurally-unrelated MCT1 and MCT4 blocker). The data are presented as the means ± standard error (SEM). (n.s.) non-significant difference. (***) indicates statistically significant differences, with *p* ≤ 0.001 using the Kruskal–Wallis test, followed by Dunn’s multiple comparisons test.

**Figure 6 metabolites-11-00432-f006:**
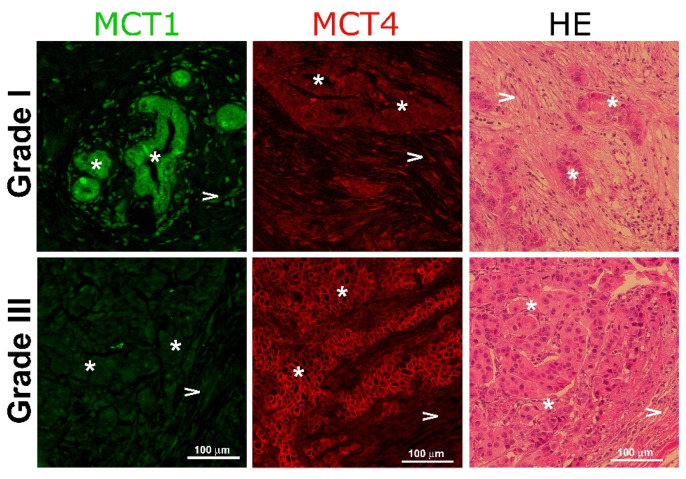
Monocarboxylate expression in human breast cancer samples. Representative immunohistochemical staining of human breast tumor samples with different Elston grades stained for MCT1 (green) and MCT4 (red), taken with a magnification of 20×. The third column represents the control stain with hematoxylin-eosin. Asterisks (*) indicate epithelial cells and arrow heads (>) show stromal cells.

## Data Availability

Data presented in this manuscript is available upon reasonable request to the corresponding author.

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
