# Peer review of "Matrix Stiffness Modulates Metabolic Interaction between Human Stromal and Breast Cancer Cells to Stimulate Epithelial Motility"

_metabolites, 2021, doi:10.3390/metabo11070432_

Round 1

Reviewer 1 Report

The manuscript by Ponce et al., reports the results of the study aimed to elucidate the influence of extracellular matrix rigidity on reprogramming of glucose metabolism in stromal cells and the effect of the produced lactate on epithelial breast cells' motility. In a set of well planed experiments, the authors show evidence that matrix stiffness stimulates stromal cells to produce lactate, which induces breast cancer cells migration. Furthermore, the involvement of MCT1 and MCT4 in this process has been examined, indirectly, by using MCT1/2 inhibitor ARC155858 and diclofenac (MCT1 & MCT4 blocker). Different response to those two inhibitors in MCF-7 and MDA-MB-231 cells, regarding cell migration stimulated by media conditioned by stromal cells, is in line with different expression of MCT1 and MCT4 as reported in ref. 18. For clarity, it could be useful to stress this difference in MCT1 and MCT4 expression in the text.

Author Response

In a previous paper (Tobar et al., ref 18), we demonstrated a flux of lactate from stromal RMF cells to breast epithelial cells. Kinetic analysis of co-cultured experiments among these two different cell types (stromal and epithelial) shown that MDA MB-231 cells are more avid for lactate (higher Vmax value) and display a Km value corresponding to MCT4. These results were consistent with the expression of both transporters in epithelial cells measured by western blot (figures 4 and 5 of the cited paper). Therefore, despite MCT4 being the primary mechanism adapted to the export of lactate, these particular malignant cells (MDA MB-231) utilize it to uptake a higher amount of lactate that, we assume, is a functional requirement to its malignancy.   

Reviewer 2 Report

Comments to the Authors

The manuscript entitled “Matrix Stiffness Modulates Metabolic Interaction between Human Stromal and Breast Cancer Cells to Stimulate Epithelial Motility” by Mr. Iván Ponce. The authors find that a stiffer extracellular matrix favors the acquisition of physical properties that promote the metabolic reprograming that also constitute a stimulus for epithelial motility. The authors proposed that matrix stiffness constitutes not only a scaffold where a tumor can develop, but also an early signal that promotes a shift of metabolic behavior of the stromal compartment through a glycolytic process that results in a stimuli of tumor malignancy. Here, I recommend accepting for publication after revisions to clarify or expand on some of the key findings of the paper.

Major comments:

  1. It is interesting that RMF-621 cells were seeded in the increased stiffness (0.2-32 kPa) and change the glucose uptake and lactate production, however if different level of stiffness (0.2-32 kPa) will affect the attachments, morphology and growth of RMF-621 cells, and then lead to switch the metabolism of glucose and lactate. 
  2. How about the protein and gene expression of MCT1, MCT4 and GLUT1 in RMF-621 cells with the increased stiffness (0.2-32 kPa)? If there are the same effects of MCT1, MCT4 and GLUT1 in RMF-621, MCF7 and MDA-231 cells with the increased stiffness treatment.
  3. In figure 5, we found doses of Diclofenac (1mM) and doses of AR–C155858 (1μM) in MCF7 and MDA-231 cells are 1,000 difference. In this case, it is hard to say that decreased effect in migrating cells with Diclofenac is solely derived from MCT4. In Discussion, explain briefly why you chose different dosages of AR-C155858 and Diclofenac.
  4. In figure 6, there is no figure staining GLUT1 and/or desmoplasia region in the human breast cancer tissues. As the authors described that desmoplasia may be a potential cause that lead to increase stiffness in breast cancer tissues, these additional figures may help the readers to clarify the relationship between fibrosis and cancer malignancy.

Minor comments:

  1. We suggest to rearrange the paragraph in the introduction, especially the line 62-64.

Author Response

MATRIX STIFFNESS MODULATES METABOLIC INTER-ACTION BETWEEN HUMAN STROMAL AND BREAST CANCER CELLS TO STIMULATE EPITHELIAL MOTILITY

Iván Ponce1, Nelson Garrido1, Nicolás Tobar1, Francisco Melo2,3, Patricio C. Smith4 and Jorge Martínez1,*

Reviewer 2.

Major comments:

  1. It is interesting that RMF-621 cells were seeded in the increased stiffness (0.2-32 kPa) and change the glucose uptake and lactate production, however if different level of stiffness (0.2-32 kPa) will affect the attachments, morphology and growth of RMF-621 cells, and then lead to switch the metabolism of glucose and lactate.

The early response of the cell to the matrix is an exciting point. During the time of contact of RMF cells with the different matrices, as expected, they attached, spread and proliferate more efficiently in the more rigid surface. However, either in the experiments where lactate production was evaluated and those in which RMF cells produce conditioned media, results were corrected by the final number of attached cells. We postulate that the whole range of functional changes induced by the interaction of stromal cells with the matrices is responsible for metabolic changes. We have no data to sustain which particular factor plays a more relevant role. These aspects are the subject of our current scientific interest.     

  1. How about the protein and gene expression of MCT1, MCT4 and GLUT1 in RMF-621 cells with the increased stiffness (0.2-32 kPa)? If there are the same effects of MCT1, MCT4 and GLUT1 in RMF-621, MCF7 and MDA-231 cells with the increased stiffness treatment.

As we mentioned, breast tumors are essentially desmoplastic entities where stromal cells are predominant. Moreover, the leading broad objective of our work is to unravel the mechanisms that allow the functional communication between the abundant stromal and the epithelial component. For this reason, our work was focused on the interaction of stromal cells with matrices of different stiffness and the consequences of this interaction in the motility behavior of epithelial cells representing two different malignant states. One of the primary purposes of our work is to highlight the mediator role of stroma in epithelial malignancy. On the other hand, the direct effect of matrix stiffness in the acquisition of malignant properties in breast epithelial cells has been widely documented. See the elegant work of Wei S.C. et al. (Nat Cell Biol. 2015;17: 678-88).

Notwithstanding the above, we had evaluated the expression of lactate transporters in epithelial cells.  These results are displayed in the figure under this text. It is important to consider that MDA MB-231 cell line express MCT4 (in a high amount) and MCT1. However, in our hands, MCF-7 cell lines express only MCT-1, been MCT4 undetectable. As figure show, different stiffness does not affect significantly the expression of either MCT4 or MCT1 in both cell lines.

Extracellular matrix stiffness does not affect the expression of monocarboxylate transporters in mammary epithelial cells. MCF-7 (A and B) and MDA MB-231 cells (C and D) were incubated on plates displaying increasing stiffness and coated with collagen I as described in Material and Methods. After 72 hours, cells were lysed and western blots to MCT1 and MCT4 was performed according to Material and Methods (A and C).  Densitometric analysis of three different experiments were performed from three independent experiments is shown in B and D. Data are presented as means ± standard error (SEM).

  1. In figure 5, we found doses of Diclofenac (1mM) and doses of AR–C155858 (1μM) in MCF7 and MDA-231 cells are 1,000 difference. In this case, it is hard to say that decreased effect in migrating cells with Diclofenac is solely derived from MCT4. In Discussion, explain briefly why you chose different dosages of AR-C155858 and Diclofenac.

We agree with the referee. Unfortunately, there are no commercially available inhibitors specific to MCT4. For this reason, we were forced to use Diclofenac that inhibits lactate transport, among other activities. The concentration of 1mM was referred to data of Contreras-Baeza Y et al (J Biol Chem. 2019; 294:20135-20147). These authors evaluate lactate transport in MDA MB-231 cells with an experimental setting similar to the one used by us.

  1. In figure 6, there is no figure staining GLUT1 and/or desmoplasia region in the human breast cancer tissues. As the authors described that desmoplasia may be a potential cause that lead to increase stiffness in breast cancer tissues, these additional figures may help the readers to clarify the relationship between fibrosis and cancer malignancy.

The purpose of this figure was only to reinforce data of figure 5 that suggest that lactate transport in the highly malignant MDA MB-231 cell line was performed by MCT4 transporter and to compare this with the expression of the transporter with a sample of an aggressive breast tumor. The idea behind this comparison is to confirm that the expression of MCT4 is a characteristic of the metastatic phenotype that it is possible to observe either in cell lines and human samples. To show a similar figure for stromal GLUT1 is a very good idea which, at this moment, is beyond our means.

Minor comments:

We suggest to rearrange the paragraph in the introduction, especially the line 62-64.

The paragraph was modified according to reviewer’s suggestion.

Reviewer 3 Report

Reciprocal interaction between breast tumors and stromal cell/cancer-associated fibroblasts contribute in the course of cancer progression in a desmoplastic environment. In the current study, the authors Ponce et al. show that increased stiffness stimulate lactate production and glucose uptake by mammary fibroblasts. Extracellular matrix stiffness modulate the expression of monocarboxylate transporter MCT4 and glucose transporter Glut-1. MCT1 and MCT4 blocker diclofenac attenuates MDA-MB-231 cell invasion. The study has been in part conducted by an elegant technique, the topic about the role of MCT4 in breast cancer is highly interesting. The result of figure 5B is highly interesting. The study could be a novel finding, but also could be a proof of concept. In a hypoxic condition, MCT4 has been shown to be upregulated. Traditionally a hypoxic condition stimulates Warburg effect and produces more lactate. The authors also mention in the discussion section about the Warburg effect.

Could you please show hypoxia marker such as HIF-1 expression in cells under different stiffness condition 0.2, 2.0, 16, and 32 KPa?

Author Response

The relationship between hypoxia and stiffed matrices has been matter of study mainly in 3D collagen matrices, system that allow to stablish hypoxic gradients that simulates microarchitecture of a solid tumors. (Cancer Res. 2019 Apr 15;79(8):1981-1995). Anyway, we had evaluated the expression of HIF-1 by qPCR in our RMF cells plated on stiffed matrices and did not find significant differences among the different stiffness (see figure above). 

Round 2

Reviewer 2 Report

The authors have addressed most of the concerns I had with the previous submission.

Reviewer 3 Report

The authors addressed the point raised by the reviewer.